# Oxidation-Induced Detachment of Ruthenoarene Units and Oxygen Insertion in Bis-Pd(II) Hexaphyrin π-Ruthenium Complexes

**DOI:** 10.3390/molecules25122753

**Published:** 2020-06-15

**Authors:** Akito Nakai, Takayuki Tanaka, Atsuhiro Osuka

**Affiliations:** Department of Chemistry, Graduate School of Science, Kyoto University, Kyoto 606-8502, Japan; nakai@shuyu.kuchem.kyoto-u.ac.jp

**Keywords:** expanded porphyrin, π-complex, ruthenium complex, aromaticity

## Abstract

Two types of new bis-Pd(II) hexaphyrin π-ruthenium complexes are reported. A double-decker bis-Pd(II) hexaphyrin π-ruthenium complex **4** was obtained by oxidation-induced detachment of a ruthenoarene unit from the triple-decker complex **3** and oxygen-inserted triple-decker bis-Pd(II) hexaphyrin π-ruthenium complex **6** was obtained upon treatment of bis-Pd(II) [26]hexaphyrin **5** with [RuCl_2_(*p*-cymene)]_2_ under aerobic conditions. Although π-metal complexation of porphyrinoids often results in decreased global aromaticity due to the enhancement of local 6π aromatic segments, distinct aromatic characters were indicated for **4** and **6** by ^1^H-NMR spectral and theoretical calculations. These results are accounted for in terms of possible resonance contributors of hexaphyrin di- and tetraanion ligands. Thus, π-metal coordination has been shown to be effective for modulation of the overall aromaticity.

## 1. Introduction

Expanded porphyrins which consist of more than four pyrrole units and methine carbons, have been attracting continuous attention due to their intriguing optical and electrochemical properties, structural diversities, and rich coordination abilities [1,2,3,4,5,6,7]. The structural and electronic characteristics of expanded porphyrins are usually quite flexible, which has been used for versatile and efficient modulation of aromaticity by redox reactions and structural modifications [5,6,7]. Recently, we report a new π-metal coordination type, wherein a (*p*-cymene)Ru^II^ fragment is η^5^-coordinated at the side pyrrole of bis-Au(III) [28]hexaphyrin, forming double-decker π-ruthenium complexes **1** and **2** (Figure 1) [8]. On the other hand, triple-decker π-ruthenium complex of bis-Pd(II) [26]hexaphyrin **3** possesses two (*p*-cymene)Ru^II^ fragments sitting on the two Pd^II^ metal centers and the inner pyrrolic β-carbon atoms in a η^6^-coordinated manner on both sides of the hexaphyrin. In the literature, π-metal complexes of porphyrinoids have been limited to those based on porphyrin, porphycene, subphthalocyanine, and subporphyrazine scaffolds [9,10,11,12,13,14,15,16], and no examples had been reported for expanded porphyrin π-metal complexes before our report [8]. Interestingly, complexes **1** and **2** showed attenuated macrocyclic paratropic ring currents, probably due to the coordination-induced attenuation of the global conjugation circuit, while complex **3** showed distinct 26π aromaticity. This difference suggested that the balance between macrocyclic aromaticity (i.e., 26/28π) and local aromaticity (i.e., 6π) is important and can be modulated by metal π-complexation [17]. This finding drove us to explore new π-ruthenium complexes of hexaphyrin. Here we report double-decker bis-Pd(II)[26]hexaphyrin π-ruthenium complex **4** and oxygen-inserted triple-decker bis-Pd(II) [26]hexaphyrin **6**. Both complexes show a distinct diatropic ring current.

## 2. Results and Discussion

Our investigation started with an attempted oxidation of triple-decker π-ruthenium complex **3** since **3** has a relatively electron-rich character (*E*_ox.1_ = 0.22 V vs Fc/Fc^+^) [8]. Therefore, oxidative titration of **3** with tris(4-bromophenyl)ammoniumyl hexachloroantimonate (TBAH) was examined. Upon addition of TBAH, **3** showed clear absorption spectral changes with isosbestic points at 426, 620, 846, and 978 nm to a spectrum with absorption maxima at 461, 624, and 799 nm (Appendix A). These spectral changes suggested ring-centered oxidation rather than metal-centered oxidation [18]. Thus we attempted to isolate the oxidized species but failed due to the instability of the oxidized species under ambient conditions. Meanwhile, we found that treatment of **3** with an excess amount of TBAH afforded a different species, **4**, in 41% yield as an entity stable under ambient conditions (Scheme 1). 

The complex **4** shows its molecular ion peak at *m*/*z* = 1901.8531 (calcd. for C_76_H_22_F_30_N_6_^106^Pd_2_^102^Ru [M]^−^: 1901.8589) by high-resolution atmospheric-pressure-chemical-ionization time-of-flight (HR-APCI-TOF) mass spectrometry, that corresponds to the double-decker complex. Indeed, treatment of [26]hexaphyrin bis-Pd(II) complex **5** [19] with 5 equivalents of [RuCl_2_(*p*-cymene)]_2_ in the presence of sodium acetate at room temperature gave the same double-decker complex **4** in 71% yield along with triple-decker complex **3** (5% yield).

The structure of **4** was determined by X-ray crystallographic analysis as shown in Figure 2. The (*p*-cymene)Ru^II^ fragment is located just above the center of the hexaphyrin framework in the same manner as the triple-decker complex **3**. The ruthenium ion (Ru1) is strongly coordinated to C2 and C4 as judged by their shorter bond lengths by about 0.1 Å than Ru1–C1 and Ru1–C3 bonds. Accordingly, the coordination of the ruthenium ion to C1, C3, Pd1 and Pd2 would be weaker similar to the case of triple-decker complex **3**. Interestingly, **4** takes a columnar stack structure and the interplanar distances between the hexaphyrin plane and *p*-cymene of another molecule were 3.441 and 3.380 Å (Appendix A).

The ^1^H-NMR spectrum of **4** showed two doublet signals due to the outer β-protons at δ = 9.45 and 9.24 ppm, indicating a distinct diatropic ring current arising from its 26π aromaticity (Figure 3). These signals were slightly downfield-shifted as compared with those of triple-decker complex **3** and parent hexaphyrin **5**, suggesting the aromaticity of **4** was enhanced. This consideration was also supported by more upfield-shifted ^1^H-NMR peaks of the *p*-cymene fragment of **4** compared with those of **3**. This enhanced aromaticity may be accounted for in terms of possible resonance contributors of the hexaphyrin ligands (vide infra). The UV/Vis/NIR absorption spectra of **3**, **4** and **5** are shown in Figure 4. In **3**, three absorption maxima are observed at 461, 650, and 829 nm, while a large absorption band at 578 nm and weak Q-like bands at 895 and 1004 nm are observed in **5**. Unlike **3** and **5**, the absorption spectrum of **4** shows more complicated bands and a weak absorption tail up to 1800 nm, and is almost insensitive to solvent polarity (Appendix A).

During the optimization of the reaction conditions, we found that a trace amount of oxygen-inserted triple-decker complex **6** was formed upon reaction of **5** with [RuCl_2_(*p*-cymene)]_2_ in air (Scheme 2). The HR-APCI-TOF MS gave the molecular ion peak of **6** at *m*/*z* = 2153.8690 (calcd. for C_86_H_36_F_30_N_6_O^106^Pd_2_^102^Ru_2_, [M]^−^: 2153.8682). The yield of **6** was improved up to 35% when the reaction was run in the presence of an excess amount (ca. 555 eq.) of water. However, the reaction in the presence of H_2_^18^O did not afford the corresponding ^18^O-incorporated product, indicating that the oxygen source might be molecular oxygen. X-Ray crystallographic analysis revealed that the oxygen atom was actually inserted between the β-carbon (C1) and Ru1 (Figure 5). The Pd1–Ru1, Pd1–Ru2, Pd2–Ru1, and Pd2–Ru2 bond lengths are 2.9548(5), 2.9651(6), 3.1146(6), and 2.8568(5) Å, respectively, all being shorter than the sums of the van der Waals radii of Pd and Ru [20]. Density functional theory (DFT) calculations [21] also indicated the weak coordinating interactions between the Ru and Pd (Appendix A). The oxygen atom (O1) is bound to Ru1 (2.120(2) Å), C1 (1.355(4) Å), and Pd1 (2.014(2) Å). Ru1 is π-coordinated to the C3-C4 bond and σ-coordinated to O1. Ru2 is π-coordinated to the C3–C4 bond, relatively strongly coordinated to C2 (2.250(4) Å) and weakly coordinated to C1 (2.396(4) Å).

On the basis of experimental results, **6** displays aromatic characters as follows: (i) The ^1^H-NMR spectrum of **6** shows eight doublet signals due to the outer β-protons in the range of δ = 8.34 − 7.88 ppm and upfield-shifted four doublets peak due to the aryl-protons of the *p*-cymene in the range of δ = 3.09 − 2.18 ppm; (ii) The nucleus-independent-chemical-shift (NICS) calculations also indicated negative values in several points inside the macrocycle of **6** (Appendix A); (iii) The absorption spectrum of **6** shows two Soret-like bands at 484 and 635 nm and one Q-like band at 879 nm (Figure 6). The lowest-energy band is relatively strong and red-shifted compared with that of **3**. Virtually no solvent effects were observed for the absorption spectra of **6** (Appendix A).

In order to understand the real electronic states, we assumed several resonance contributors in **3**, **4**, and **6** as shown in Scheme 3. For **4**, two resonance contributors can be considered, both of which have 26π conjugation circuits regardless of the involvement of anions. As is the case in freebase hexaphyrins [22,23] and hexaphyrin bis-Pd(II) complex [19], the aromaticity of hexaphyrin dianion species becomes stronger than that of the neutral species. In contrast, the tetraanion of the hexaphyrin core in **3** can be drawn with three resonance contributors, one of which has a 28π antiaromatic circuit. In order to estimate the relative contributions, the harmonic oscillator stabilization energies (HOSE) [24] have been calculated based on the crystal structures. The estimated weights for the canonical structures **3A**, **3B**, and **3C** are 0.342, 0.334 and 0.325, respectively. The calculated electrostatic potential map of **3** displayed relatively electronegative positions being roughly consistent with these canonical structures (Appendix A). Therefore, it could be assumed that non-negligible antiaromatic contribution caused the outer β-protons of **3** slightly upfield-shifted in the ^1^H-NMR spectrum as compared with **4**, in which both of the 26π resonance contributors are estimated to be almost equally contributed (0.481 and 0.519 for **4A** and **4B**, respectively). With regard to **6**, six resonance contributors are considered as shown in Scheme 3c. The estimated weights for the canonical structures **6A** to **6F** can be estimated to be 0.189, 0.182, 0.168, 0.162, 0.152, and 0.147, respectively. Similarly to the case of **3**, the resonance contributors with 26π systems (**6A** and **6B**) are more important than those with 28π systems (**6E** and **6F**).

The electrochemical properties of **4** and **6** were examined by cyclic voltammetry (Table 1). [26]Hexaphyrin bis-Pd^II^ complex **5** showed a relatively large electrochemical HOMO–LUMO gap (1.21 eV) as a typical feature of an aromatic hexaphyrin [25]. Four reversible waves at 0.81, 0.56, −0.43, and −1.01 V versus a ferrocene/ferrocenium ion couple were observed for double-decker complex **4** in CH_2_Cl_2_. While the electrochemical HOMO–LUMO gap was also increased upon Ru^II^ metalation from **5** (1.21 eV) to **3** (1.51 eV), the oxidation and reduction potentials of **4** were negatively and positively shifted, respectively, in comparison with those of **5**, thus giving rise to a smaller gap of 0.99 eV in **4**. The oxidation and reduction potentials of **6** were slightly negatively and positively shifted, respectively, from those of **3**, thus giving rise to a smaller gap of 1.25 eV. The order of the electrochemical HOMO–LUMO gaps is in accordance with the spectral red-shifts at their lowest-energy bands. In addition, DFT calculations indicated the HOMO–LUMO gaps to be 1.76, 1.40, and 1.62 eV, for **3** [8], **4**, and **6**, respectively (Appendix A). The tendency of calculated HOMO–LUMO energy gaps also reflect their π-metalation-dependent aromaticity modulations.

## 3. Materials and Methods

The chemicals used for synthesis were of reagent grade quality unless otherwise mentioned. Dry toluene was obtained by distillation over CaH_2_. Dry DMF was purchased from Wako (FUJIFILM Wako Pure Chemical Corportaion, 1-2, Doshomachi 3-chome, Chuo-ku, Osaka, Japan). Silica gel column chromatography was performed on Wako gel C-300 and C-400 unless otherwise mentioned. ^1^H (600.17 MHz) and ^19^F (564.73 MHz) NMR spectra were acquired on an ECA-600 spectrometer (JEOL Ltd., 3-1-2 Musashino, Akishima-shi, Tokyo, JAPAN) and chemical shifts were reported as the delta scale in ppm relative to CHCl_3_ as internal references for ^1^H (δ = 7.26 ppm), and hexafluorobenzene as an external reference for ^19^F (δ = −162.9 ppm). Coupling constants (*J*) are given in Hz. High-resolution atmospheric-pressure-chemical-ionization time-of-flight (HR-APCI-TOF) mass spectra were recorded on a micrOTOF LC instrument BRUKER Daltonics (Bruker Corporation, 3-9, Moriya-cho, Kanagawa-ku, Yokohama-shi, Kanagawa, Japan). UV/Vis/NIR absorption spectra were recorded on a UV-3600PC spectrometer (Shimadzu Corporation, 1, Nishinokyo Kuwabara-cho, Nakagyo-ku, Kyoto, Japan). Redox potentials were measured by cyclic voltammetry on model ALS 612E electrochemical analyzer (BAS Inc., 1-28-12, Mukaijima, Sumida-ku, Tokyo, Japan). X-Ray single crystal diffraction analyses were performed on an XtaLAB P200 apparatus (Rigaku Corporation, 3-9-12, Matsubara-cho, Akishima-shi, Tokyo, Japan) at −180 °C using a two-dimensional PILATUS 100K/R detector with Cu*K*α radiation (λ = 1.54187 Å). The structures were solved by direct method SHELEXT-2014/5, and refined by SHELEXL-2014/7 programs [26,27,28].

### 3.1. Oxidation of ***3*** into [26]Hexaphyrin Bis-Pd(II)-Mono-Ru(II) Complex ***4***

A solution containing **3** (11.6 mg, 5.5 μmol) and tris(4-bromophenyl)ammoniumyl hexachloroantimonate (202.1 mg, 248 μmol, 45 eq.) in CH_2_Cl_2_ was stirred for 10 min. The resulting mixture was passed through γ-alumina column and the solvent was removed under reduced pressure. The residue was separated by silica-gel column chromatography (C300, CH_2_Cl_2_:*n*-hexane = 1:2 to 1:0) and a red fraction was collected. Recrystallization from CH_2_Cl_2_/*n*-hexane afforded **4** as dark red solids (3.6 mg, 2.2 μmol, 41%).

### 3.2. Direct Synthesis of [26]Hexaphyrin Bis-Pd(II)-Mono-Ru(II) Complex ***4***

A solution containing [26]hexaphyrin bis-Pd(II) complex **5** (162.2 mg, 97.2 μmol), [RuCl_2_(*p*-cymene)]_2_ (304.9 mg, 498 μmol, 5 eq.), and sodium acetate (84.0 mg, 1.00 mmol, 10 eq.) in dry toluene (40 mL) and dry DMF (10 mL) was stirred for 13 h at room temperature. The solution was added to water and extracted with CH_2_Cl_2_. The combined organic layer was dried over Na_2_SO_4_ and the solvent was removed in vacuo. The crude mixture was separated by silica-gel column chromatography (CH_2_Cl_2_:*n*-hexane = 1:2 to 1:0), and a green fraction and a red fraction were collected. Removal of solvent of green fraction afforded **3** as green solids (10.9 mg, 5.1 μmol, 5%). Recrystallization of red fraction from CH_2_Cl_2_ / *n*-hexane afforded **4** as dark red solids (130.4 mg, 68.6 μmol, 71%). Compound data of **3** were described [8]. Compound data of **4** are as follows: ^1^H-NMR (CDCl_3_, 298 K) δ [ppm] = 9.45 (d, 4H, *J* = 4.6 Hz, β-H), 9.24 (d, 4H, *J* = 4.6 Hz, β-H), 0.16 (d, 2H, *J* = 5.5 Hz, Ar-H), 0.13 (d, 2H, *J* = 5.5 Hz, Ar-H), −1.77 (d, 6H, *J* = 6.9 Hz, CH(C*H*_3_)_2_), −2.01 (s, 3H, Me), and −2.38 (m, 1H, C*H*(CH_3_)_2_). ^19^F-NMR (CDCl_3_, 298 K) δ [ppm] = −133.97 (d, 4F, *J* = 21.7 Hz, *o*-F), −135.90 (d, 2F, *J* = 30.3 Hz, *o*-F), −136.67 (d, 2F, *J* = 17.3 Hz, *o*-F), −139.65 (d, 4F, *J* = 17.3 Hz, *o*-F), −151.07 (t, 2F, *J* = 21.7 Hz, *p*-F), −153.98 (t, 4F, *J* = 21.7 Hz, *p*-F), −160.97 (t, 4F, *J* = 21.7 Hz, *m*-F), −161.63 (t, 8F, *J* = 19.5 Hz, *m*-F), −162.59 (t, 5F, *J* = 19.5 Hz, *m*-F), and −163.33 (t, 4F, *J* = 21.7 Hz, *m*-F). UV/Vis/NIR (CH_2_Cl_2_): λ_max_ [nm] (ε [M^−1^cm^−1^]) = 345 (29000), 410 (26000), 487 (45000), 548 (43000), 677 (51000), 812 (17000), and 1179 (1600).

HR APCI-TOF-MS (negative): *m*/*z* calcd. for C_76_H_22_F_30_N_6_^106^Pd_2_^102^Ru: 1901.8589, [*M*]^−^; found: 1901.8531.

### 3.3. [26]Hexaphyrin Bis-Pd(II)-Bis-Ru(II)-Oxygen Inserted Complex ***6***

A solution containing **5** (17.2 mg, 10.3 μmol), [RuCl_2_(*p*-cymene)]_2_ (32.0 mg, 52.3 μmol, 5 eq.), and sodium acetate (8.3 mg, 98.8 mmol, 9 eq.) in toluene (4 mL), DMF (1 mL) and H_2_O (0.1 mL) was stirred for 12 h at 100 °C. The solution was added to water and extracted with CH_2_Cl_2_. The combined organic layer was dried over Na_2_SO_4_ and the solvent was removed in vacuo. The crude mixture was separated by silica-gel column chromatography (C-400, CH_2_Cl_2_:*n*-hexane = 1:2), and a green fraction and a dark green fraction were collected. Removal of solvent of the green fraction afforded **3** as green solids (9.2 mg, 4.3 μmol, 42%). Removal of solvent of the dark green fraction afforded **6** as green solids (7.8 mg, 3.6 μmol, 35%). ^1^H-NMR (CDCl_3_, 298 K) δ [ppm] = 8.34 (d, 1H, *J* = 5.5 Hz, β-H), 8.24 (d, 1H, *J* = 4.1 Hz, β-H), 8.20 (d, 1H, *J* = 4.6 Hz, β-H), 8.17 (d, 1H, *J* = 4.6 Hz, *β*-H), 8.13 (d, 1H, *J* = 4.6 Hz, β-H), 8.08 (d, 1H, *J* = 5.5 Hz, β-H), 8.01 (d, 1H, *J* = 4.6 Hz, β-H), 7.88 (d, 1H, *J* = 5.0 Hz, β-H), 3.09 (d, 2H, *J* = 5.5 Hz, Ar-H), 2.79 (d, 2H, *J* = 5.5 Hz, Ar-H), 2.46 (d, 2H, *J* = 5.5 Hz, Ar-H), 2.18 (d, 2H, *J* = 5.5 Hz, Ar-H), 1.25 (s, 6H, CH(C*H*_3_)_2_), 0.86 (m, 1H, C*H*(CH_3_)_2_), −0.01 (m, 6H, CH(C*H*_3_)_2_), −0.29 (s, 3H, Me), −0.59 (s, 3H, Me), and −1.87 (m, 1H, C*H*(CH_3_)_2_). ^19^F-NMR (CDCl_3_, 298 K) δ [ppm] = −135.03 (s, 1F, *o*-F), −135.34 (s, 1F, *o*-F), −136.11 (d, 1F, *J* = 21.7 Hz, *o*-F), −136.90 (s, 1F, *o*-F), −137.28 (d, 2F, *J* = 17.3 Hz, *o*-F), −137.80 (d, 2F, *J* = 26.0 Hz, *o*-F), −138.46 (d, 2F, *J* = 17.3 Hz, *o*-F), −139.33 (d, 1F, *J* = 21.7 Hz, *o*-F), −139.66 (d, 1F, *J* = 26.0 Hz, *o*-F), −152.58 (t, 1F, *J* = 21.7 Hz, *p*-F), −152.78 (t, 1F, *J* = 19.5 Hz, *p*-F), −154.49 (t, 1F, *J* = 19.5 Hz, *p*-F), −155.09 (t, 1F, *J* = 21.7 Hz, *p*-F), −155.52 (m, 2F, *p*-F), −161.76 (m, 4F, *m*-F), −162.85 (t, 1F, *J* = 19.5 Hz, *m*-F), −163.18 (t, 1F, *J* = 19.5 Hz, *m*-F), −163.59 (t, 1F, *J* = 19.5 Hz, *m*-F), and −164.00(m, 5F, *m*-F). UV/Vis/NIR (CH_2_Cl_2_): λ_max_ [nm] (*ε* [M^−1^cm^−1^]) = 353 (31000), 484 (58000), 635 (38000), and 879 (21000). HR APCI-TOF-MS (negative): *m*/*z* calcd. for C_86_H_36_F_30_N_6_O^106^Pd_2_^102^Ru_2_: 2153.8682, [*M*]^−^; found: 2153.8690.

### 3.4. Synthesis of [26]Hexaphyrin Bis-Pd(II)-Bis-Ru(II)-Oxygen Inserted Complex ***6*** under Argon Atmosphere

A solution containing **5** (16.7 mg, 10.0 μmol), [RuCl_2_(*p*-cymene)]_2_ (30.9 mg, 50.5 μmol, 5 eq.), and sodium acetate (9.2 mg, 109 μmol, 10 eq.) in dry toluene (4 mL) and dry DMF (1 mL) was stirred for 9 h at 100 °C under argon atmosphere. After the reaction mixture was cooled to room temperature, the solution was added to water and extracted with CH_2_Cl_2_. The combined organic layer was dried over Na_2_SO_4_ and the solvent was removed in *vacuo*. The crude mixture was separated by silica-gel column chromatography (C-400, CH_2_Cl_2_:*n*-hexane = 1:2), and the green fraction was collected. Removal of solvent afforded **3** as green solids (14.6 mg, 6.8 μmol, 68%). Only a trace amount of **6** was detected.

## 4. Conclusions

In summary, one of the ruthenoarene units of the triple-decker complex **3** was detached by oxidation with TBAH, affording the new double-decker complex **4**. While the aromaticity of **3** was weaker than that of parent [26]hexaphyrin **5**, that of **4** was slightly enhanced. This conflicting result was explained by the resonance contributors of hexaphyrin di- and tetraanion ligands. Given the fact that π-complexation often disturbs the macrocyclic ring current, **4** is a quite rare example whose aromaticity was enhanced by π-metal coordination. For oxygen-inserted triple-decker complex **6**, one of the two rutheniums was π-coordinated to a C=C double bond on one side and was σ-coordinated to oxygen on the other side. Again, this complex exhibited distinct aromatic characters, suggesting the active involvement of 26π aromatic resonance contributors. Further studies on novel π-complexes of expanded porphyrins are actively ongoing.

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
