# Peer review of "Oxidation-Induced Detachment of Ruthenoarene Units and Oxygen Insertion in Bis-Pd(II) Hexaphyrin π-Ruthenium Complexes"

_molecules, 2020, doi:10.3390/molecules25122753_

Round 1

Reviewer 1 Report

This was a very focussed but interesting paper describing the specific transformation of a bimetallic complex with an extended pi system, it properties and its reaction with a ruthenium aryl complex to form an addition complex.

An impressive range of methods were employed in order to study the novel complexes, most notably X-ray crystallography. In addition NMR data was obtained, absorption spectra and of course extensive DFT was carried out in order to understand the multiple resonance forms of the complex ligand. An intermediate in the oxidation process - containing an additional O atom, was also obtained and characterised by X-ray crystalligraphy.

The paper is well-written and interesting to read. The immediate application is not obvious but that does not matter - this is high quality basic research of good quality and worthy of publication.

Author Response

We appreciate the reviewer's high evaluations and positive comments. As the reviewer mentioned, the immediate application is not obvious, but we will try to expend this motif to other expanded porphyrins with expectation to observe novel coordination modes as well as conceptually novel (anti)aromatic systems.

Reviewer 2 Report

molecules-831793

The manuscript submitted by Osuka et al. described the synthesis of two types of new bis-Pd(II) hexaphyrin π-ruthenium complexes and their characterization. This is a good paper and can be published after only very minor revision.

Some detail suggestions:

  1. The captions for Figure 3 should give the meaning of * and should distinguish the resonances of methyl and methine of iPr.
  2. Page 4, line 93. One more reaction is required to confirm the possibility of oxygen source of compound 6. By running the reaction under nitrogen with the same condition as the paper described on page 9, line212~214. If compound 6 is still isolated, the oxygen source is from the water. Otherwise, the author’s assumption in page 4, line 93 is correct.

Author Response

We appreciate the reviewer's high evaluations and suggestions for improvement.

>>>The captions for Figure 3 should give the meaning of * and should distinguish the resonances of methyl and methine of iPr.

We modified Figure 3 to distinguish the resonances of iPr. Asterisks mean the peaks assignable to residual solvents and impurity.

>>>Page 4, line 93. One more reaction is required to confirm the possibility of oxygen source of compound 6. By running the reaction under nitrogen with the same condition as the paper described on page 9, line212~214. If compound 6 is still isolated, the oxygen source is from the water. Otherwise, the author’s assumption in page 4, line 93 is correct. 

We appreciate the reviewer's thoughtful suggestions. We run the same reaction under argon atmosphere and observed only trace amount of oxygenated compound 6. The experimental detail is shown in the revised manuscript. 

Reviewer 3 Report

The authors have synthesized two new bis-Pd(II) hexaphyrin ruthenium complexes. The molecules are well characterized using different spectroscopies and mass-spectrometry. They have also performed calculations to get deeper insights. However, the authors could do a much better job on the computational side. Cyclic voltammograms yielded oxidation and reduction potentials. They claim that the HOMO-LUMO gap is obtained by adding the absolute values of them. The sum of the ionization potential and the electron affinity is not the HOMO-LUMO gap. The electron affinity is the ionization potential of the anion or the HOMO energy of the anion, if one assumes that Koopmans' theorem is fulfilled. This discussion has to be reformulated. Spectroscopical studies suggest that molecules 4 and 6 are aromatic. The authors investigate the resonance structures to understand the aromaticity. In Scheme 3, two structures have been assumed for 4. The ACID plot in the SI suggests that the ring current passes on the inside and via the beta carbons of the pyrrole rings. The resonance structures do not allow that possibility implying that the interpretation, the model or the way of thinking is incorrect. Many calculations on porphyrins show that ring currents usually divide into a inner and outer branch at the pyrroles. This seems to be the case also for 4 according to the ACID plots. Six resonance structures of molecule 6 was suggested. Two of them are antiaromatic. Mixture of aromatic and antiaromatic resonance structures is like mixing oil and water. The ACID plot in the SI suggests that little current takes the outer branch at the pyrrole rings. The dominating pathway is the inner one in the 4 corners, whereas the current is stronger along the outer branch at the pyrrole rings in the middle. That aromatic pathway is excluded in the study. Judged from the thickness of the aromatic pathway in the ACID plots, I would guess that 4 has a stronger ring current than 6. NISC(0) values suggest the opposite. However, NICS(0) values are more or less random numbers, which has been documented in many studies. I would not draw conclusions based on them. The authors use small basis sets and ECPs that I would not recommend. The basis sets and ECPs are used in many studies, but that does not mean that they are good. Some details that I found: in the abstract complexation on should be complexation of on line 47 is a relatively should be has a relatively on line 49 clear absorption spectral changes with clear isosbestic. clear appear twice I would remove one of them. In the SI I found possition which should be position.

Author Response

We appreciate the reviewer's critical comments.

>>>Cyclic voltammograms yielded oxidation and reduction potentials. They claim that the HOMO-LUMO gap is obtained by adding the absolute values of them. The sum of the ionization potential and the electron affinity is not the HOMO-LUMO gap. The electron affinity is the ionization potential of the anion or the HOMO energy of the anion, if one assumes that Koopmans' theorem is fulfilled. This discussion has to be reformulated. 

We determined the oxidation potentials by calculating the midpoint of two potentials; one is oxidation of neutral, the other is reduction of cation species, as is often deduced from sweeped voltammetric curves. The same method is applied for reduction potentials. Thus, the obtained "electrochemical" H-L gap is useful measure for comparison and we use this value as usual. Since we agree with the reviewer's claim, we add the term "electrochemical" HOMO-LUMO gap in the revised manuscript.

>>>The authors investigate the resonance structures to understand the aromaticity. In Scheme 3, two structures have been assumed for 4. The ACID plot in the SI suggests that the ring current passes on the inside and via the beta carbons of the pyrrole rings. The resonance structures do not allow that possibility implying that the interpretation, the model or the way of thinking is incorrect. Many calculations on porphyrins show that ring currents usually divide into a inner and outer branch at the pyrroles. This seems to be the case also for 4 according to the ACID plots. Six resonance structures of molecule 6 was suggested. Two of them are antiaromatic. Mixture of aromatic and antiaromatic resonance structures is like mixing oil and water. The ACID plot in the SI suggests that little current takes the outer branch at the pyrrole rings. The dominating pathway is the inner one in the 4 corners, whereas the current is stronger along the outer branch at the pyrrole rings in the middle. That aromatic pathway is excluded in the study. Judged from the thickness of the aromatic pathway in the ACID plots, I would guess that 4 has a stronger ring current than 6. NISC(0) values suggest the opposite. However, NICS(0) values are more or less random numbers, which has been documented in many studies. I would not draw conclusions based on them.

The suggested aromatic pathway has been actually considered for HOSE estimations, but the contribution is quite weak. This is shown in the revised manuscript as resonance structures 4A' and 4B'.  We thank the reviewer's thoughtful ideas for computations, but these are just a support for experimental observations. Inconsistency may be attributed to poor basis sets for calculations. The point is replied as shown below. 

>>> The authors use small basis sets and ECPs that I would not recommend. The basis sets and ECPs are used in many studies, but that does not mean that they are good. 

We agree with the reviewer's concern. However, because the molecule is large in size (and hence the number of valuables are quite large), we could not do calculations with better functions. As shown in Table S1 and S2, the bond lengths are roughly consistent with observed structures. From the experience in our past works, removing C6F5 groups caused serious discrepancies so that we cannot omit them.

>>>in the abstract complexation on should be complexation of 

Corrected.

>>> on line 47 "is a relatively" should be "has a relatively"

Corrected.

>>> on line 49 clear absorption spectral changes with clear isosbestic. clear appear twice I would remove one of them.

Corrected.

>>>In the SI I found possition which should be position.

Corrected.